# DIFFUSION MODELS WITHOUT ATTENTION

## ABSTRACT

Advances in high-fidelity image generation have been spearheaded by denoising diffusion probabilistic models (DDPMs). However, there remain considerable computational challenges when scaling current DDPM architectures to high-resolutions, due to the use of attention either in UNet architectures or Transformer variants. To make models tractable, it is common to employ lossy compression techniques in hidden space, such as patchifying, which trade-off representational capacity for efficiency. We propose Diffusion State Space Model (DiffuSSMs), an architecture that replaces attention with a more efficient state space model backbone. The model avoids global compression which enables longer, more fine-grained image representation in the diffusion process. Comprehensive validation on ImageNet indicates superior performance in terms of FiD and Inception Score at reduced total FLOP usage compared to previous diffusion models using attention.

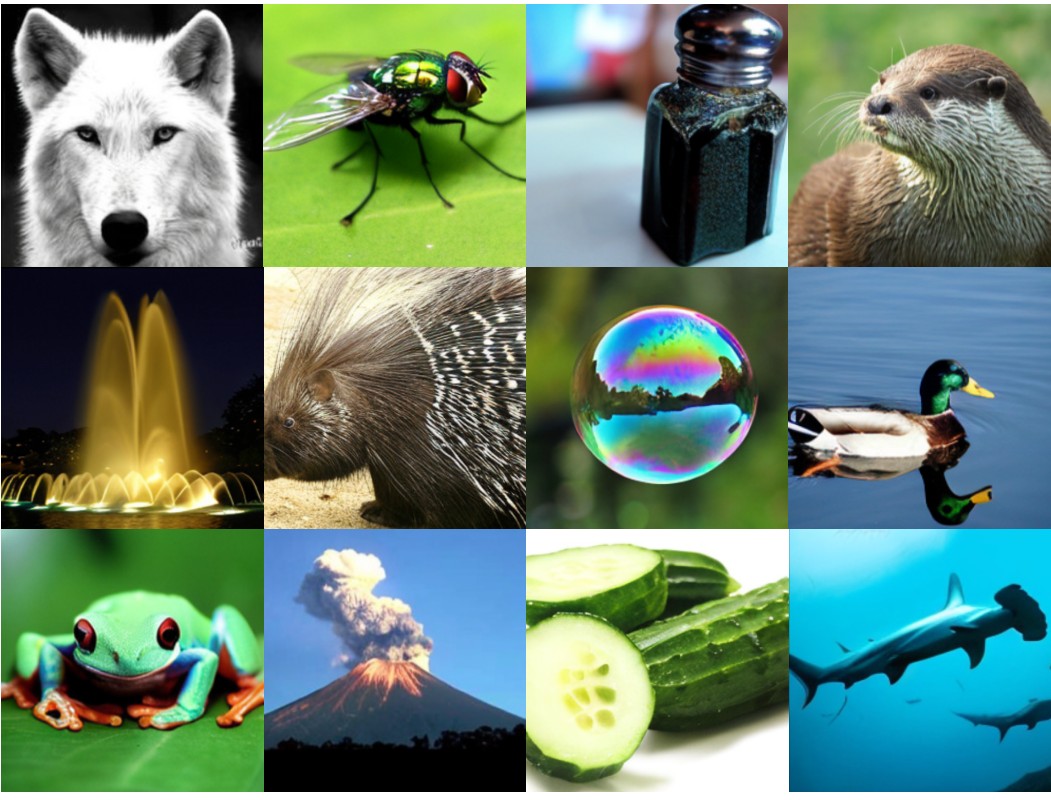

Figure 1: Example images generated by class-conditional DIFFUSSM at $256 \times 256$ resolution.

## 1 INTRODUCTION

Rapid progress in image generation has been driven by denoising diffusion probabilistic models (DDPMs) (Ho et al., 2020; Nichol & Dhariwal, 2021; Dhariwal & Nichol, 2021). DDPMs pose

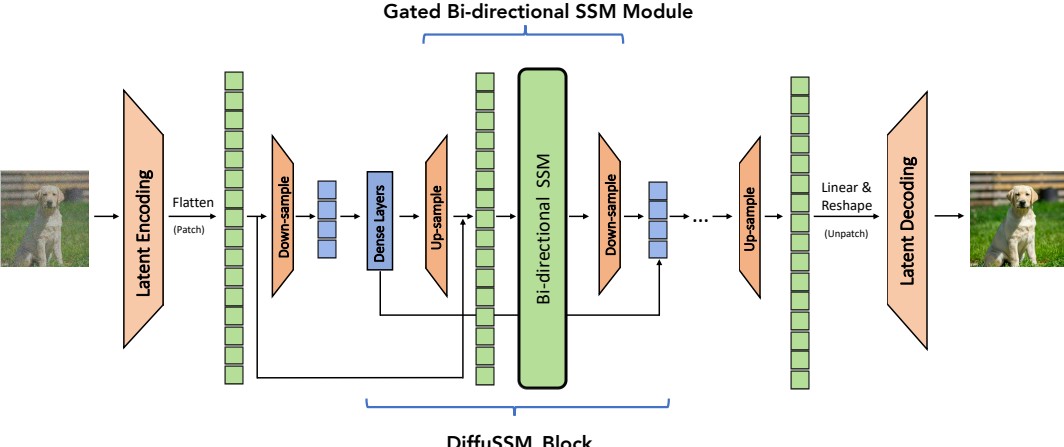

Figure 2: Architecture of DIFFUSSM. DIFFUSSM takes a noised image representation, flattens it to a sequence, and applies repeated layers alternating long-range SSM cores with hour-glass feed-forward networks. Unlike with U-Nets or Transformers there is no application of patchification or scaling for the long-range block.

the generative process as iteratively denoising latent variables, yielding high-fidelity samples when enough denoising steps are taken. Their ability to capture complex visual distributions makes DDPMs promising for advancing high-resolution, photorealistic synthesis.

However, significant computational challenges remain in scaling DDPMs to higher resolutions. A major bottleneck is the reliance on the attention mechanism for high-fidelity generation. In convolutional U-Nets architectures, this bottleneck comes from combining ResNet (He et al., 2016) with attention layers (Van den Oord et al., 2016; Salimans et al., 2017). DDPMs surpass generative adversarial networks (GANs) with the inclusion of multi-head attention layers (Vaswani et al., 2017) as shown by Nichol & Dhariwal (2021); Dhariwal & Nichol (2021). In Transformer architectures (Vaswani et al., 2017), attention is the central component, and is therefore critical for achieving recent state-of-the-art image synthesis results (Peebles & Xie, 2022; Bao et al., 2023). In both these architectures, the complexity of attention becomes prohibitive when working with high-resolution images.

Computational costs have motivated the use of representation compression methods. High-resolution architectures generally employ patchifying (Bao et al., 2023; Peebles & Xie, 2022), or multi-scale resolution(Ho et al., 2020; Nichol & Dhariwal, 2021; Hoogeboom et al., 2023). Patchifying creates coarse-grained representations which reduces computation at the cost of degraded critical high-frequency spatial information and structural integrity (Peebles & Xie, 2022; Bao et al., 2023; Schramowski et al., 2023). Multi-scale resolution, while alleviating computation at attention layers, can diminish spatial details through downsampling (Zamir et al., 2021) and can introduce artifacts (Wang et al., 2020b) while applying up-sampling.

The Diffusion State Space Model (DIFFUSSM), is an attention-free diffusion architecture, shown in Figure 2, that aims to circumvent the issues of applying attention for high-resolution image synthesis. DIFFUSSM utilizes a gated state space model (SSM) backbone in the diffusion process. Previous work has shown that sequence models based on SSMs are an effective and efficient general-purpose neural sequence model (Gu et al., 2021). By using this architecture, we can enable the SSM core to process finer-grained image representations by removing global patchification or multi-scale layers. To further improve efficiency, DIFFUSSM employs an hourglass architecture for the dense components of the network. Together these approaches target the asymptotic complexity of length as well as the practical efficiency in the position-wise portion of the network.

We validate DIFFUSSM's across different resolutions. Experiments on ImageNet demonstrate consistent improvements in FiD, sFiD, and Inception Score over existing approaches in various resolutions with fewer total Gflops.

## 2 BACKGROUND

### 2.1 DIFFUSION MODELS

Denoising Diffusion Probabilistic Models (DDPMs) (Ho et al., 2020) are generative models that sample from a complex, high-dimensional distribution. They simulate a stochastic process where an initial image $x_0$ is gradually corrupted by noise, transforming it into a simpler, noise-dominated state. This forward noising process can be represented as follows:

$$q(x_{1:T}|x_0) = \prod_{t=1}^{T} q(x_t|x_{t-1}), \ \ q(x_t|x_0) = N(x_t; \sqrt{\bar{\alpha}_t}x_0, (1 - \bar{\alpha}_t)I), \tag{1}$$

where $x_{1:T}$ denotes a sequence of noised images from time $t = 1$ to $t = T$. The main task of the DDPM is to learn the *reverse* process that recovers the original image utilizing learned $\mu_\theta$ and $\Sigma_\theta$:

$$p_\theta(x_{t-1}|x_t) = N(x_{t-1}; \mu_\theta(x_t), \Sigma_\theta(x_t)), \tag{2}$$

Our main focus will be on the parameters $\theta$. These parameters $\theta$ are trained to maximize the variational lower bound on the log-likelihood of the observed data $x_0$: $\max_\theta -\log p_\theta(x_0|x_1) + \sum_t D_{KL}(q^*(x_{t-1}|x_t, x_0) \,||\, p_\theta(x_{t-1}|x_t))$. To simplify the training process, researchers reparameterize $\mu_\theta$ as a function of the predicted noise $\varepsilon_\theta$ and minimize the mean squared error between $\varepsilon_\theta(x_t)$ and the true Gaussian noise $\varepsilon_t$: $\min_\theta ||\varepsilon_\theta(x_t) - \varepsilon_t||_2^2$. However, in order to train a diffusion model that can learn a variable reverse process covariance $\Sigma_\theta$, we need to optimize the full $L$. We follow the approach of Peebles & Xie (2022) to train the network: we use the simple objective to train the noise prediction network $\varepsilon_\theta$ and use the full objective to train the covariance prediction network $\Sigma_\theta$.

### 2.2 ARCHITECTURES FOR DIFFUSION MODELS

To model $\mu_\theta(x_t)$, it is necessary to learn a function from noised images to images. It is common to work with images in latent space, where the image is first transformed by an encoder and then decoded back to its original shape (Rombach et al., 2022). We consider methods for parameterizing the function $\mathbb{R}^{H \times W \times C} \to \mathbb{R}^{H \times W \times C}$ where $H, W, C$ are the height, width, and size of the latent space representation. When generating high-resolution images, even in latent space, $H$ and $W$ are large, and require specialized architectures for this function to be tractable.

**UNets with Multiscale Architectures** U-Net architectures (Ho et al., 2020; Nichol & Dhariwal, 2021; Hoogeboom et al., 2023) use multi-scale resolution to handle high-resolution images. Let $t_1, \dots t_T$ be a series of lower-resolution feature maps created by down-sampling the image.[1] At each scale a ResNet (He et al., 2016) is applied to $\mathbb{R}^{H_t \times W_t \times C_t}$. These are then upsampled and combine into the final output. To enhance the performance of U-Net in image generation, attention layers are integrated at the lowest-resolutions. The feature map is flattened to a sequence of $H_t W_t$ vectors. For instance, when considering $H = 256 \times W = 256$ down to attention layers of $16 \times 16$ and $32 \times 32$, leading to sequences of length 256 and 1024 respectively. Applying attention earlier improves accuracy at a larger computational cost.

**Transformers with Patchification** Transformer architectures utilize attention throughout, but handle high-resolution images through patchification (Dosovitskiy et al., 2020). Given a patch size $P$, the transformer partitions the image into $P \times P$ patches yielding a new $\mathbb{R}^{H/P \times W/P \times C'}$ representation. This patch size $P$ directly influences the effective granularity of the image and downstream computational demands. To feed patches into a Transformer, the image is flattened and a linear embedding layer is applied to obtain a sequence of $(HW)/P^2$ hidden vectors (Peebles & Xie, 2022; Bao et al., 2023; Hoogeboom et al., 2023; Dosovitskiy et al., 2020). Due to this embedding step, which projects from $C'$ to the model size, large patches risk loss of spatial details and ineffectively model local relationships due to reduced overlap. However, patchification has the benefit of reducing the quadratic cost of attention as well as the feed-forward networks in the Transformer.

---

[1]Note that choices of up- and down-sample include learned parameters and non-parameterized ones such as average pooling and upscale (Hoogeboom et al., 2023; Croitoru et al., 2023).

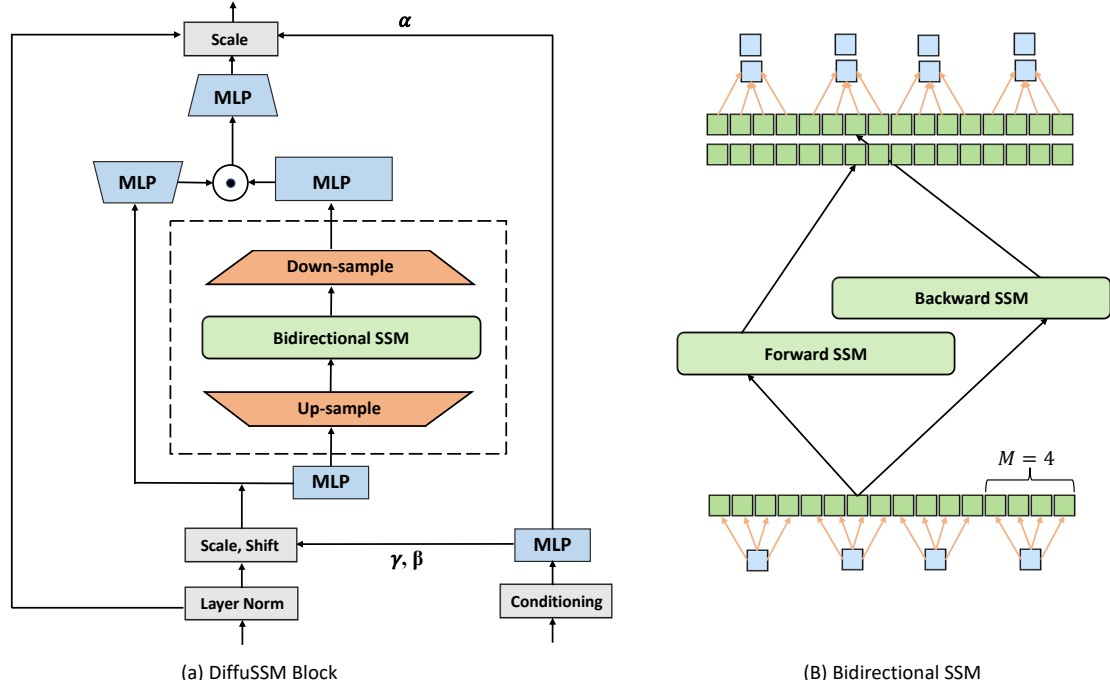

Figure 3: Details of the DIFFUSSM Block. (a) Model architecture of one single layer with four central MLPs ($\mathbf{W}^1, \mathbf{W}^2, \mathbf{W}^3, \mathbf{W}^4$) and gating, (b) Detailed illustration of the Bidirectional SSM and parameterized up/down-sampling ($\mathbf{W}^\downarrow, \mathbf{W}^\uparrow$ w/ $M = 4$).

## 3  SSMs FOR DIFFUSION WITHOUT ATTENTION

Our goal is to produce a diffusion architecture that learns long-range interactions at high-resolution without requiring attention layers and patchification. As with Transformer models for diffusion, the approach works by flattening the image and treating it like a sequence modeling problem.

### 3.1  STATE SPACE MODELS

SSM neural networks are a class of architectures for processing discrete-time sequences (Gu et al., 2021). The models behave like a linear recurrent neural network (RNN) processing an input sequence of scalars $u_1, \ldots u_L$ to output $y_1, \ldots y_L$ with the following equation,

$$x_k = \overline{\boldsymbol{A}}x_{k-1} + \overline{\boldsymbol{B}}u_k, \quad y_k = \overline{\boldsymbol{C}}x_k.$$

Where $\overline{\boldsymbol{A}} \in \mathbb{R}^{N \times N}, \overline{\boldsymbol{B}} \in \mathbb{R}^{N \times 1}, \overline{\boldsymbol{C}} \in \mathbb{R}^{1 \times N}$. The main benefit of this approach, compared to alternative architectures such as Transformers and standard RNNs, is that the linear structure allows it to be implemented using a long *convolution* as opposed to a recurrence. Specifically, $y$ can be computed from $u$ with an FFT yielding $O(L \log L)$ complexity, allowing it to be applied to significantly longer sequences. When handling vector inputs, we can stack $D$ different SSMs and apply a $D$ batched FFTs.

However a linear RNN, by itself, is not an effective sequence model. The key insight from past work is that if the discrete-time values $\overline{\boldsymbol{A}}, \overline{\boldsymbol{B}}, \overline{\boldsymbol{C}}$ are derived from appropriate continuous-time state-space models, the linear RNN approach can be made stable and effective (Gu et al., 2020). We therefore learn a continuous-time SSM parameterization $\boldsymbol{A}, \boldsymbol{B}, \boldsymbol{C}$ as well as a discretization rate $\Delta$, which is used to produce the necessary discrete-time parameters. Original versions of this conversion were challenging to implement, however recently researchers (Gu et al., 2022; Gupta et al., 2022) have introduced simplified diagonalized versions of SSM neural networks that achieve comparable results with a simple approximation of the continuous-time parameterization. We use one of these, S4D (Gu et al., 2022), as our backbone model.

Just as with RNNs, SSMs can be made bidirectional by concatenating the outputs of two SSM layers and passing them through an MLP to yield a $L \times 2D$ output. In addition, past work shows that

this layer can be combined with multiplicative gating to produce an improved Bidirectional SSM layer (Wang et al., 2022; Mehta et al., 2022) as part of the encoder, which is the motivation for our architecture.

## 3.2 DIFFUSSM BLOCK

The core block of DIFFUSSM is a gated bidirectional SSM. The SSM is designed to reduce the cost of working with long sequences. To further reduce the cost of the MLP layers, we use an hour-glass architecture that expands and contracts the length of the sequence around the SSM layer. The full block is shown in Figure 3.

Specifically, each layer receives a shortened, flattened input sequence $\mathbf{I} \in \mathbb{R}^{J \times D}$ where $M = L/J$ is the downsampling ratio. We compute the following for $l \in \{1 \dots L\}$ with $j = \lfloor l/M \rfloor, m = l \bmod M, D_m = 2D/M$.

$$
\begin{aligned}
\mathbf{U}_l &= \sigma(\mathbf{W}_k^\uparrow \sigma(\mathbf{W}^0 \mathbf{I}_j)) && \in \mathbb{R}^{L \times D} \\
\mathbf{Y} &= \text{Bidirectional-SSM}(\mathbf{U}) && \in \mathbb{R}^{L \times 2D} \\
\mathbf{I}'_{j, D_m k : D_m (k+1)} &= \sigma(\mathbf{W}_k^\downarrow \mathbf{Y}_l) && \in \mathbb{R}^{J \times 2D} \\
\mathbf{O}_j &= \mathbf{W}^3(\sigma(\mathbf{W}^2 \mathbf{I}'_j) \odot \sigma(\mathbf{W}^1 \mathbf{I}_j)) && \in \mathbb{R}^{J \times D}
\end{aligned}
$$

We integrate this Gated SSM block in each layer. Additionally, following past work we integrate a combination of the class label $\mathbf{y} \in \mathbb{R}^{L \times 1}$ and timestep $\mathbf{t} \in \mathbb{R}^{L \times 1}$ at each position, as illustrated.

**Parameters and FLOPs**   The number of parameters in the DIFFUSSM block is dominated by the linear transforms, $\mathbf{W}$, these contain $9D^2 + 2MD^2$ parameters. With $M = 2$ this yields $13D^2$ parameters. The DiT transformer block has $12D^2$ parameters in its core transformer layer; however the DiT architecture has more parameters in other layer components (adaptive layer norm). We match parameters in experiments by using an additional DIFFUSSM layer.

The total Flops in one layer of DIFFUSSM is $13\frac{L}{M}D^2 + LD^2 + \alpha 2 L \log LD$ where $\alpha$ represents a constant for the FFT implementation. With $M = 2$ and noting that the linear layers dominate computation, this yields roughly $7.5LD^2$ Gflops. In comparison, if instead of using SSM, we had used self-attention, we would have $DL^2$ additional Flops. Considering our two experimental scenarios: 1) $D \approx L = 1024$ which would have given $LD^2$ extra Flops, 2) $4D \approx L = 4096$ which would give $4LD^2$ Flops and significantly increase cost. DiT avoids these issues by using patching as discussed earlier, at the cost of representational compression.

## 4 EXPERIMENTAL STUDIES

### 4.1 EXPERIMENTAL SETUP

Our primary experiments are conducted on the ImageNet dataset (Deng et al., 2009) at $256 \times 256$ and $512 \times 512$ resolution. We used latent space encoding which gives effective sizes $32 \times 32$ and $64 \times 64$ with $L = 1024$ and $L = 4096$ respectively.

**Linear Decoding and Weight Initialization**   After the final block of the Gated SSM, the model decodes the sequential image representation to the original spatial dimensions to output noise prediction and diagonal covariance prediction. Similar to Peebles & Xie (2022); Gao et al. (2023), we use a linear decoder and then rearrange the representations to obtain the original dimensionality. We follow DiT to use the standard layer initializations approach from ViTDosovitskiy et al. (2020).

**Training Configuration**   We followed the same training recipe from DiT (Peebles & Xie, 2022) to maintain an identical setting across all models. We also chose to follow existing literature to keep an exponential moving average (EMA) of model weights with a constant decay. Off-the-shelf VAE encoders from [2] were used, with parameters fixed during training. Our DIFFUSSM-XL possesses

---

[2]https://github.com/CompVis/stable-diffusion

approximately 673M parameters and encompasses 29 layers of Bidirectional Gated SSM blocks with a model size $D = 1152$. This value is similar to DiT-XL. trained our model using a mixed-precision training approach to mitigate computational costs. We adhere to the identical configuration of diffusion as outlined in ADM (Dhariwal & Nichol, 2021), including their linear variance scheduling, time and class label embeddings, as well as their parameterization of covariance $\Sigma_\theta$. More details can be found in the Appendix.

**Metrics**    To quantify the performance of image generation of our model, we used Frechet Inception Distance(FID) (Heusel et al., 2017), a common metric measuring the quality of generated images. We followed convention when comparing against prior works and reported FID-50K using 250 DDPM sampling steps. We also reported sFID score (Nash et al., 2021), which is designed to be more robust to spatial distortions in the generated images. For a more comprehensive insight, we also presented the Inception Score (Salimans et al., 2016) and Precision/Recall (Kynkäänniemi et al., 2019) as supplementary metrics. Note that do not incorporate classifier-free guidance unless explicitly mentioned(we used $-G$ for the usage of classifier-free guidance or explicitly state the CFG).

**Implementation and Hardware**    We implemented all models in Pytorch and trained them using NVIDIA A100. DIFFUSSM-XL, our most compute-intensive model trains on 8 A100 GPUs 80GB with a global batch size of 256. More computation details and speed can be found in the Appendix.

## 4.2    BASELINES

We compare to a set of previous best models, these include: GAN-style approaches that previously achieved state-of-the-art results, UNet-architectures trained with pixel space representations, and Transformers operating in the latent space. More details can be found in Table 4.3. Our aim is to compare, through a similar denoising process, the performance of our model with respect to other baselines. Some recent studies Zheng et al. (2023); Gao et al. (2023) focusing on image generation at the $256 \times 256$ resolution level have combined masked token prediction with existing DDPM training objectives to advance the state of the art. However, these works are orthogonal to our primary comparison, so we have not included them in Table 1.

## 4.3    EXPERIMENTAL RESULTS

$256 \times 256$ **Benchmark**    We compare DIFFUSSM with state-of-the-art class-conditional generative models, as depicted in Table 4.3. When classifier-free guidance is not employed, DIFFUSSM outperforms other diffusion models in both FID and sFID, reducing the best score from the previous non-classifier-free latent diffusion models from 9.62 to 9.22, while utilizing $3.25\times$ fewer training steps.[3] In terms of Total Gflops of training, our uncompressed model yields a $29\%$ reduction of the total Gflops compared with DiT. When classifier-free guidance is incorporated, our models attain the best sFID score among all DDPM-based models, exceeding other state-of-the-art strategies, demonstrating the images generated by DIFFUSSM are more robust to spatial distortion. As for FID score, DIFFUSSM surpasses all other models except DiT when using classifier-free guidance with which there is a 0.31 gap.

$512 \times 512$ **Benchmark**    We further compare on a higher-resolution benchmark using CFG. Results from DIFFUSSM here are relatively strong and near some of the state-of-the-art high-resolution models, beating all models but DiT on sFID and achieving comparable FID scores. The DIFFUSSM was trained on 192M images, seeing 1/4 as many images and using half the Gflops as DiT.

Table 1: Image generation quality evaluation of DIFFUSSM and existing approaches(BigGAN-deep (Brock et al., 2018), MaskGIT (Chang et al., 2022), StyleGAN-XL (Sauer et al., 2022), ADM (Dhariwal & Nichol, 2021), LDM-8 (Rombach et al., 2022), DiT-XL/2 (Peebles & Xie, 2022), CDM (Ho et al., 2022), U-Vit (Bao et al., 2023)) on ImageNet 256× 256. We took reported results from other papers with their # trained images. We calculated the Total images by training steps × batch size as reported in their paper, and total Gflops by Total Images × GFlops/Image.

**ImageNet 256×256 Benchmark**

| Models | Total Images(M) | Total Gflops | FID ↓ | sFID ↓ | IS ↑ | Pr ↑ | Re ↑ |
|---|---|---|---|---|---|---|---|
| BigGAN-deep | - | - | 6.95 | 7.36 | 171.40 | 0.87 | 0.28 |
| MaskGIT | 355 | - | 6.18 | - | 182.1 | 0.80 | 0.51 |
| StyleGAN-XL | - | - | **2.30** | 4.02 | 265.12 | 0.78 | 0.53 |
| ADM | 507 | $5.68 \times 10^{12}$ | 10.94 | 6.02 | 100.98 | 0.69 | 0.63 |
| ADM-U | 507 | $3.76 \times 10^{11}$ | 7.49 | 5.13 | 127.49 | 0.72 | 0.63 |
| CDM | - | - | 4.88 | - | 158.71 | - | - |
| LDM-8 | 307 | $1.75 \times 10^{10}$ | 15.51 | - | 79.03 | 0.65 | 0.63 |
| LDM-4 | 213 | $2.22 \times 10^{10}$ | 10.56 | - | 103.49 | 0.71 | 0.62 |
| DiT-XL/2 | 602 | $7.10 \times 10^{10}$ | 10.67 | - | - | - | - |
| DiT-XL/2 | 1792 | $2.13 \times 10^{11}$ | 9.62 | 6.85 | 121.50 | 0.67 | 0.67 |
| **DIFFUSSM-XL** | 338 | $9.57 \times 10^{10}$ | 9.86 | 5.51 | 111.59 | 0.69 | 0.64 |
| **DIFFUSSM-XL** | 550 | $1.52 \times 10^{11}$ | **9.22** | 5.64 | 111.46 | 0.69 | 0.65 |
| CFG | | | | | | | |
| ADM-G | 507 | $5.68 \times 10^{11}$ | 4.59 | 5.25 | 186.70 | 0.82 | 0.52 |
| ADM-G, ADM-U | 507 | $3.76 \times 10^{12}$ | 3.60 | - | 247.67 | 0.87 | 0.48 |
| LDM-8-G | 307 | $1.75 \times 10^{10}$ | 7.76 | - | 209.52 | 0.84 | 0.35 |
| LDM-4-G | 213 | $2.22 \times 10^{10}$ | 3.95 | - | 178.2 2 | 0.81 | 0.55 |
| U-ViT-H/2-G | 512 | $6.81 \times 10^{10}$ | 2.29 | - | 247.67 | 0.87 | 0.48 |
| DiT-XL/2(G-1.25) | 1792 | $2.13 \times 10^{11}$ | 3.22 | 5.28 | 201.77 | 0.76 | 0.62 |
| DiT-XL/2(G-1.5) | 1792 | $2.13 \times 10^{11}$ | 2.27 | 4.60 | 278.24 | 0.83 | 0.57 |
| **DIFFUSSM-XL**(G-1.5) | 338 | $9.57 \times 10^{10}$ | 2.91 | 4.55 | 247.84 | 0.76 | 0.62 |
| **DIFFUSSM-XL**(G-1.5) | 550 | $1.52 \times 10^{11}$ | 2.58 | **4.53** | 249.72 | 0.85 | 0.55 |
| **ImageNet 512×512 Benchmark** | | | | | | | |
| ADM | 1385 | $5.97 \times 10^{11}$ | 23.24 | 10.19 | 58.06 | 0.73 | 0.60 |
| ADM-U | 1385 | $3.9 \times 10^{12}$ | 9.96 | 5.62 | 121.78 | 0.75 | 0.64 |
| ADM-G | 1385 | $5.97 \times 10^{11}$ | 7.72 | 6.57 | 172.71 | 0.87 | 0.42 |
| ADM-G, ADM-U | 1385 | $4.5 \times 10^{12}$ | 3.85 | 5.86 | 221.72 | 0.84 | 0.53 |
| U-ViT/2-G | 512 | $6.81 \times 10^{10}$ | 4.05 | 8.44 | 261.13 | 0.84 | 0.48 |
| DiT-XL/2-G | 768 | $4.03 \times 10^{11}$ | **3.04** | **5.02** | 240.82 | 0.84 | 0.54 |
| **DIFFUSSM-XL-G** | 192 | $2.03 \times 10^{11}$ | 4.41 | 5.40 | 208.64 | 0.87 | 0.51 |

## 5 ANALYSIS

### 5.1 EFFICIENCY ANALYSIS

Computational cost is core to scaling diffusion models. Training models with DIFFUSSM leads to different Gflops / steps tradeoff when compared to patchification. We therefore compare theoretical Gflops and empirical training speed of several variant models. To make empirical comparisons, we utilize the implementation from the official DiT repo[4]. We remove additional optimization techniques

---

[3]Due to computational expense limits, our model was not able to be trained with a similar scale of images as DiT, which contains $7000K$ images. We also were not able to search for optimal CFG hyperparameters and report with values from past work.

[4]https://github.com/facebookresearch/DiT

Table 2: (Left) Ablation experiment showing training curve with different sampling measures. (Right) Efficiency analysis of DIFFUSSM and DiT under different resolution and patch size settings.

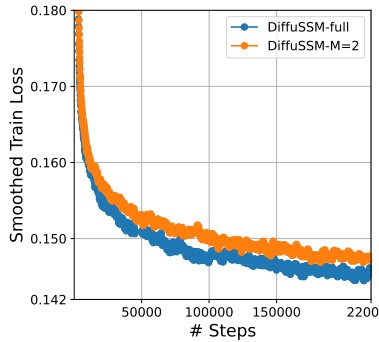

| | Effective Length | Gflops/ Image | Images/ Second |
|---|---|---|---|
| DiT-XL | 1024 | 507.95 | 86.30 |
| DiT-XL/2 | 256 | 118.64 | 396.40 |
| DiffuSSM-XL (fp32) | 1024 | 285.02 | 199.61 |
| DiffuSSM-XL | 1024 | 142.50 | 310.30 |
| DiT-XL | 4096 | 2452.16 | 13.36 |
| DiT-XL/2 | 1024 | 524.60 | 70.93 |
| DiffuSSM-XL (fp32) | 4096 | 1087.50 | 43.72 |
| DiffuSSM-XL | 4096 | 559.72 | 77.52 |

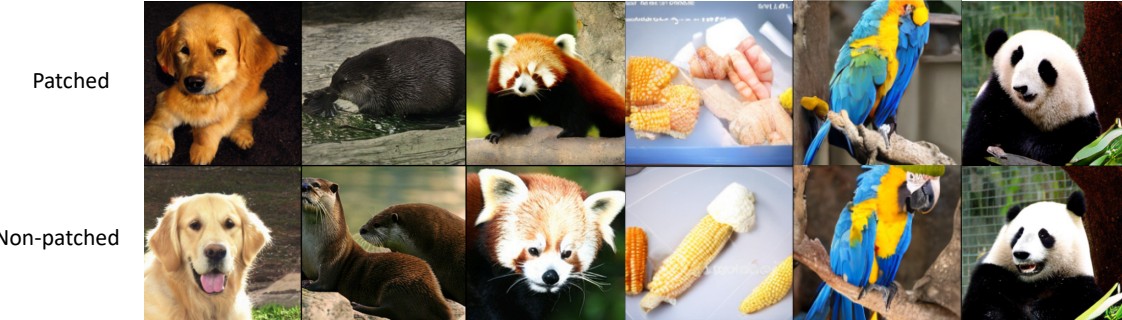

Figure 4: Qualititative Studies of Patching and Non-Patching of DIFFUSSM.

from both models, such as gradient checkpointing, and maximize the utilization of GPU memory to determine the batch size.

Results are displayed in Table reftab:efficiency (Right). We note that for all models, runtime roughly matches their predicted Gflops. Additionally while DiT=XL/2 with patching is more efficient per step, DiT without patching is much slower to use the same effective length and at longer lengths the gap grows. DIFFUSSM is able to expand to longer lengths while keeping a reasonable computational cost. For both models, further system optimization techniques, like FlashAttention (Dao et al., 2022a) for transformers and FlashConv (Fu et al., 2023) for SSMs may also reduce the run-time overhead.

## 5.2 ABLATION STUDIES

**Impact of Hourglass**   We trained our model with different sampling settings to assess the impact of compression in latent space: using a downsampling ratio $M = 2$ (our regular model), and one with $M = 1$. We have applied a smooth window to the average to better reveal the tendency. We found that the loss curve is a good performance indicator at the first 250k steps. Hence we report the loss curve of the first 250k steps as shown in Table 2(left). While $M = 2$ impacts the loss convergences, we found that the Gflop efficiency makes up for a small degradation.

**Qualitative Analysis**   The objective of DIFFUSSM is to avoid compressing hidden representations. To test whether this is beneficial we compare to a version of DIFFUSSM that uses patchifying DIFFUSSM-XL/2. We trained this variant and the DIFFUSSM-XL architecture for 400K training steps using the same starting noise and noise schedule for each class label. Figure 4 shows the results of the two model. We note that by removing the patching, the generated images at the same training steps are more robust to the spatial reconstruction yielding better visual quality.

## 6   RELATED WORK

**Diffusion Models**   Denoising Diffusion Probabilistic Models (DDPMs) (Ho et al., 2020; Nichol & Dhariwal, 2021; Sohl-Dickstein et al., 2015; Hoogeboom et al., 2023) are an advancement in the diffusion models family. Previously, Generative Adversarial Networks (GANs) (Goodfellow et al., 2014) were preferred for generation tasks. Diffusion and score-based generative models (Song et al., 2020b) have shown considerable improvements, especially in image generation tasks (Rombach et al., 2022; Saharia et al., 2022; Ramesh et al., 2022). Key enhancements in DDPMs have been largely driven by improved sampling methodologies (Ho et al., 2020; Nichol & Dhariwal, 2021; Karras et al., 2022), and the incorporation of classifier-free guidance (Ho & Salimans, 2022). Additionally, Song et al. (2020a) has proposed a faster sampling procedure known as Denoising Diffusion Implicit Model(DDIM). Latent space modeling is another core technique in deep generative models. Variational autoencoders (VAEs) (Kingma & Welling, 2013) pioneered learning latent spaces with encoder-decoder architectures for reconstruction. A similar compression idea was applied in diffusion models as the recent Latent Diffusion Models (LDMs) (Rombach et al., 2022) held state-of-the-art sample quality by training deep generative models to invert a noise corruption process in a latent space when it was first proposed. Additionally, recent approaches also developed masked training procedures, augmenting the denoising training objectives with masked token reconstruction (Gao et al., 2023; Zheng et al., 2023). Our work is fundamentally built upon existing DDPMs, particularly the classifier-free guidance paradigm.

**Architectures for Diffusion Models**   Early diffusion models utilized U-Net style architectures (Ho et al., 2020) inspired by image segmentation models. Subsequent works enhanced U-Nets with techniques like more layers of attention layers at multi-scale resolution level (Dhariwal & Nichol, 2021; Nichol & Dhariwal, 2021), residual connections (Brock et al., 2018), and conditional normalization (). However, U-Nets face challenges in scaling to high resolutions due to the growing computational costs of the attention mechanism (Shaham et al., 2018). Recently, vision transformers (ViT) (Dosovitskiy et al., 2020) have emerged as an alternate architecture given their strong scaling properties and long-range modeling capabilities proving that convolution inductive bias is not always necessary. Diffusion transformers (Peebles & Xie, 2022; Bao et al., 2023) demonstrated promising results. Other hybrid CNN-transformer architectures were proposed (Liu et al., 2021) to improve training stability. Our work aligns with the exploration of sequence models and related design choices to generate high-quality images but focuses on a complete attention-free architecture.

**Efficient Long Range Sequence Architectures**   The standard transformer architecture employs self-attention to comprehend the interaction of each individual token within a sequence. However, it encounters challenges when modeling extensive sequences due to the quadratic computational requirement. Several attention approximation methods (Wang et al., 2020a; Ma et al., 2021; Tay et al., 2020; Shen et al., 2021; Hua et al., 2022) have been introduced to approximate self-attention within sub-quadratic space. Mega(Ma et al., 2022) combines exponential moving average with a simplified attention unit, surpassing the performance of transformer baselines. Venturing beyond the traditional transformer architectures, researchers are also exploring alternate models adept at handling elongated sequences. State space models (SSM)-based architectures(Gu et al., 2021; Gupta et al., 2022; Gu et al., 2022) have yielded significant advancements over contemporary state-of-the-art methods on the LRA benchmark. Furthermore, Dao et al. (2022b); Poli et al. (2023); Peng et al. (2023) have substantiated the potential of non-attention architectures in attaining commendable performance in language modeling. Our work draws inspiration from this evolving trend of diverting from attention-centric designs and predominantly utilizes the backbone of the SSM model family.

## 7   CONCLUSION

We introduce DIFFUSSM, an architecture for diffusion models that does not require the use of Attention. This approach can handle long-ranged hidden states without requiring representation compression. Results show that architecture can achieve better performance than DiT models utilizing less Gflops at 256x256 and competitive results at higher-resolution even with less training. This model provides an alternative approach for learning effective diffusion models at large scale. Additionally removing the attention bottleneck should open up the possibility of applications in other areas that requires fine-grained diffusion, for example high-fidelity audio, video, or 3D modeling.

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

## APPENDIX

### DETAILED CONFIGURATION OF MODEL TRAINING

All our models are trained using AdamW (Loshchilov & Hutter, 2017; Kingma & Ba, 2014) optimizer. To have a comparison with DiT, we set the learning rate to be $1e - 4$ in all experiments that we have conducted. Meanwhile, to avoid other random factors, we don't do any weight decay. Additionally, we keep the batch size to be identical to DiT paper to avoid potential effects brought by a larger batch size. Accumulative Gradients were used to facilitate this goal. We also applied an Exponential Moving Average of DiT weights during our training with the decay factor to be 0.9999. We didn't search for the best configuration

