# OpenReview forum: "Diffusion Models without Attention"
_ICLR.cc/2024/Conference — ICLR 2024 Conference Withdrawn Submission_

### Official Review · Reviewer_kiL5 · 2023-10-29

**Soundness:** 2 fair
**Presentation:** 3 good
**Contribution:** 2 fair
**Rating:** 3
**Confidence:** 4

**Summary:**

Denoising diffusion probabilistic models (DDPMs) have advanced high-fidelity image generation, but they struggle with high resolutions due to attention mechanisms. The paper introduces Diffusion State Space Model (DiffuSSMs) to replace attention with a more efficient approach. DiffuSSMs excel in fine-grained image representation, outperforming some previous models in FiD and Inception Score on ImageNet with reduced computational load.

**Strengths:**

1. The rationale for substituting the attention mechanism with a more efficient network structure to alleviate computational burdens in high-resolution image generation is sound.
2. Furthermore, the proposed method demonstrates commendable performance on 256x256 images.

**Weaknesses:**

1. The article's motivation is to enhance the accuracy and computational efficiency of high-resolution image generation. However, the results on the ImageNet 512×512 Benchmark in Table 1 reveal a performance degradation of the proposed method compared to DiT-XL/2-G, along with increased computational resource consumption relative to U-ViT/2-G. Notably, the absence of comparisons with state-of-the-art methods for 512x512 resolution, such as Stable Diffusion, raises questions. To substantiate the claim that the proposed method is suitable for high-resolution generation, additional comparisons at higher resolutions, like 1024x1024, should be included.
2. It appears that the author has transplanted the network structure from Bidirectional Gated SSM without adequately explaining the task-specific adaptations and improvements. This lack of clarity can be perceived as a limitation in terms of contributions to the research community.
3. The ablation studies presented a lack of rigor. Table 2 primarily focuses on changes in loss without directly demonstrating the efficiency gains and quality degradation resulting from "COMPRESSION IN LATENT SPACE." This makes it challenging to assess the true effectiveness of this component. Additionally, relying solely on visual representations may be insufficient, especially given the stochastic nature of the diffusion model.

**Questions:**

1. In the ImageNet 256×256 Benchmark, both ADM-U and CDM exhibit lower FID scores than the proposed method. This raises questions about the emphasis placed on the proposed method's performance.
2. The presence of two DIFFUSSM-XL (or DIFFUSSM-XL(G-1.5)) entries in the ImageNet 256×256 Benchmark necessitates clarification regarding the distinctions between these two models.
3. The presentation of Table 2 raises concerns due to the inconsistent use of bold formatting for metrics, which can be confusing for readers.

---

### Official Review · Reviewer_gSqZ · 2023-10-29

**Soundness:** 2 fair
**Presentation:** 2 fair
**Contribution:** 2 fair
**Rating:** 3
**Confidence:** 4

**Summary:**

This paper explores the network architecture design for diffusion model, aiming to balance the representation capacity and efficiency. Specifically, it introduces the state space model to the network and reduce the computational burden.  The experiment results verify the effectiveness of the designed network to some extent.

**Strengths:**

According to the results shown in the paper, the proposed network is able to generate realistic images with rich structure.

**Weaknesses:**

1. The writing of this paper is very bad. Firstly, it does not clearly explain the proposed architecture. I have read the method part twice and still can't capture the detailed architecture. Second, there are a lot of typos in this paper.

2. I suggest to mark the numbers for the equations. I'm not an expert for SSM, so I can't understand the meaning for the notation $x_k$ in the first equation of Sec. 3.1. Please give specific explanation for each notation appeared.

3. Please mark $W^1$ to $W^4$ in Fig. 3. The mathematical formulation in Sec. 3.2 can not corresponds well with Fig. 3.

4. Please check the citation for the table number in Sec. 4.3

5. In Table 1, what's difference between ADM-U and ADM-G? what's the difference between "G-1.25" and "G-1.5"?

6. Table 1 can not illustrate the superiority of the proposed method.

**Questions:**

See weakness.

---

### Official Review · Reviewer_PFc7 · 2023-10-30

**Soundness:** 3 good
**Presentation:** 3 good
**Contribution:** 2 fair
**Rating:** 3
**Confidence:** 4

**Summary:**

This paper combines SSM with diffusion models and proposes DiffuSSMs to improve the efficiency in diffusion models. Instead of using attention layers in the diffusion models as most previous works does, this paper replaces the attention layers with a more efficient state space model backbone. The paper conducts experiments on ImageNet 256x256 and 512x512 benchmarks and also provides efficiency analysis.

**Strengths:**

1. The overall method is straightforward by replacing the attention layer with SSM to enable longer, more fine-grained image representation.
2. The scaling problem exists in the diffusion models, and the proposed method seems a feasible way to alleviate this problem.

**Weaknesses:**

1. Though the main motivation is that the current diffusion models are hard to scale to high-resolutions, the paper only shows results on 256x256 and 512x512 benchmarks (both utilizes latent space encoding which is actually 32x32 and 64x64) and does not show its performance on high-resolution images such as 1k images.
2. The overall performance looks similar when utilizing CFG on 256x256 benchmarks and does not show advantages on 512x512 benchmarks.

**Questions:**

N/A.

---

### Official Review · Reviewer_Rpnw · 2023-10-30

**Soundness:** 2 fair
**Presentation:** 2 fair
**Contribution:** 2 fair
**Rating:** 3
**Confidence:** 3

**Summary:**

This submission deals with designing efficient architecture for diffusion models to replace the computationally expensive self- attention layers. As an alternative to capture long-range dependencies, state-space models are used. The idea is to flatten the input image and treat it as a sequence generation problem, where downsampling and upsampling layers are used along with a bidirectional SSM. Experiments with imagenet dataset report superior FID and inception score at reduced FLOPS.

**Strengths:**

Designing more efficient architecture for diffusion model is a timely problem

**Weaknesses:**

The motivation for replacing attention layers is not very clear. Diffusion UNets typically have only a few attention layers at lower feature resolutions. The authors need to motivate if that is the computational bottleneck for training or sampling?

The reported experiments are not sufficient to make a conclusion about the effectiveness of the method at different scales. Different network sizes are needed, e.g., a plot of FID versus the network size for different architectures to see how this new architecture scales. Also, this architecture needs to be tested with other datasets. Tests with CIFAR and LAION would be very useful to convince the readership.

More ablations are needed to compare with simple baseline s such as the diffusion UNet after removing the attention layers.

**Questions:**

The displayed generated images e.g., in Fig. 1 and Fig. 4 look smooth. The fine details are missing? Is this the artifact of the new architecture?

The motivation behind choosing the DiffuSSM block as in section 3.2 is not discussed. More intuition is needed. What’s the reason to use downsampling before SSM. is it just for compression purposes?